# Multi-scale chromatin state annotation using a hierarchical hidden Markov model

Eugenio Marco[1],*, Wouter Meuleman[2],*, Jialiang Huang[1],*, Kimberly Glass[3], Luca Pinello[1], Jianrong Wang[2], Manolis Kellis[2] & Guo-Cheng Yuan[1]

Chromatin-state analysis is widely applied in the studies of development and diseases. However, existing methods operate at a single length scale, and therefore cannot distinguish large domains from isolated elements of the same type. To overcome this limitation, we present a hierarchical hidden Markov model, diHMM, to systematically annotate chromatin states at multiple length scales. We apply diHMM to analyse a public ChIP-seq data set. diHMM not only accurately captures nucleosome-level information, but identifies domain-level states that vary in nucleosome-level state composition, spatial distribution and functionality. The domain-level states recapitulate known patterns such as super-enhancers, bivalent promoters and Polycomb repressed regions, and identify additional patterns whose biological functions are not yet characterized. By integrating chromatin-state information with gene expression and Hi-C data, we identify context-dependent functions of nucleosome-level states. Thus, diHMM provides a powerful tool for investigating the role of higher-order chromatin structure in gene regulation.

[1] Department of Biostatistics and Computational Biology, Dana-Farber Cancer Institute and Harvard T.H. Chan School of Public Health, Boston, Massachusetts 02215, USA. [2] Computer Science and Artificial Intelligence Laboratory, Massachusetts Institute of Technology and Broad Institute, Cambridge, Massachusetts 02139, USA. [3] Channing Division of Network Medicine, Department of Medicine, Brigham and Women's Hospital and Harvard Medical School, Boston, Massachusetts 02215, USA. * These authors contributed equally to this work. Correspondence and requests for materials should be addressed to M.K. (email: manoli@mit.edu) or to G.-C.Y. (email: gcyuan@jimmy.harvard.edu).

More than a decade since the completion of the Human Genome Project[1], our understanding of genome function remains incomplete. One of the main reasons is that, although the majority of the genome does not code for genes, many noncoding regions have important regulatory functions[2,3]. This is mechanistically achieved in part by packing the genome into chromatin, whose cell-type-specific states reflect the accessibility of transcriptional factors and their proximity to target genes. At the basic level, chromatin structure contains multidimensional nucleosome structural information along the single-dimension genomic coordinates. To elucidate the biological role of these basic structures, several computational analysis tools have been developed to systematically classify nucleosome-level chromatin states[4–6]. These tools have been very successful in the discovery and annotation of millions of regulatory regions, such as enhancers and promoters, in various cell types[7–10]. However, they have been unable to unravel higher-order chromatin structures.

Chromatin forms higher-order three-dimensional structures by folding and looping[11], facilitating long-range interactions between enhancers and target genes[12]. While the factors determining such long-range interactions remain poorly understood, the process is likely related to the distribution of histone marks over broad domains[13–15]. Recently, the identification of broad domains has drawn considerable interest[13,14,16,17], and a number of computational methods in the literature can be used to segment chromatin at large scales. For example, in Graph-Based Regularization, Libbrecht et al.[18] combine a chromatin-state segmentation algorithm with Hi-C data, with the underlying idea that regions of the genome that are in close physical proximity will share the same chromatin-state annotation. However, this method is only applicable to cell types for which high-resolution Hi-C data are available that is still a stringent constraint due to the technical difficulty and formidable cost of Hi-C experiments. Knijnenburg et al.[19] developed a multiscale approach to visualize and analyse genomic signals; however, this method is limited to analysing a single genomic feature at a time. Chen et al.[20] developed a multivariate Bayesian change point (BCP) model to identify break points of broad chromatin domains that they called BLOCKs; however, this method does not provide information about the biological function of BLOCKs.

To systematically annotate the chromatin states at multiple length scales, we have developed a new computational method called hierarchical hidden Markov model (diHMM). Our method not only inherits the advantage of ChromHMM in integrating multiple chromatin data sets and discovering reoccurring combinatorial and spatial patterns de novo, but further extends by providing a modelling framework that systematically identifies combinatorial patterns at multiple length scales, thereby enabling the detection of latent domain states and their associations with nucleosome-scale chromatin states.

## Results

**diHMM is a hierarchical hidden Markov model.** diHMM differs from existing methods in that it uses a hierarchical hidden Markov model framework, where each level of hidden states corresponds to a distinct length-scale (Fig. 1). It can be used to analyse any number of levels of chromatin states (Methods). diHMM takes multiple ChIP-seq (chromatin immunoprecipitation with sequencing) data as input, and outputs a genome-wide segmentation of the genome into functionally annotated, multi-level chromatin states, each corresponding to a specific length scale.

For simplicity, we focus on a two-level model (see Methods for discussion regarding extension to incorporate additional layers),

where the lower level corresponds to nucleosome-level states and the upper level corresponds to broader domain-level states (Fig. 1a and Supplementary Fig. 1). Following the approach taken by ChromHMM[21], we first binarize each data track at a 200-base pair (bp) resolution, approximately the size of a nucleosome. The combinatorial patterns of chromatin marks at the 200 bp bins are classified by a discrete set of nucleosome-level states. Domain-level states are used to annotate the transition patterns between nucleosome-level states over regions covered by 20 consecutive 200 bp bins and thus have a 4 kb resolution. At each genomic locus, the assignment of domain-level and nucleosome-level states is interdependent: with domain states informing the overall frequency of different nucleosome states, whereas nucleosome-level states over multiple 200 bp bins provide the transitional grammar for domain-level state classification. These two levels of chromatin states can be identified simultaneously using an iterative algorithm (see Methods for details). For functional analysis, we consider the combination of both levels of chromatin states. By using a relatively small number of states in each level, diHMM can effectively capture a large number of combinatorial patterns.

We applied diHMM to annotate multi-scale chromatin states in the three ENCODE tier 1 cell lines, H1 (human embryonic stem cells), GM12878 (B cell-derived lymphoblastoid cells) and K562 (erythroleukemia cells), using a public ChIP-seq data set containing 9 marks: CTCF, H3K4me3, H3K4me2, H3K4me1, H3K9ac, H3K27ac, H3K36me3, H4K20me1 and H3K27me3 (ref. 2). Following previous studies[7,10], we determined the number of chromatin states based on a balance between biological complexity, model interpretability and speed. As a result, we constructed a model containing 30 nucleosome-level and 30 domain-level states. As discussed later, the results are not significantly affected by the number of chromatin states. diHMM provides genome-wide annotations of chromatin states. However, due to the lack of numerical efficiency, it is infeasible to train a diHMM model using genome-wide data. Therefore, we selected a short chromosome (chromosome 17) as training set, combining information from all three cell lines. The model was then applied to annotate the entire genome. To test the robustness of diHMM, we retrained a model based on data from chromosome 20. The results are in good agreement (Supplementary Fig. 2). Compared with the nucleosome-level states, the domain-level states are less robust, likely reflecting the smaller sample size in the training data. In addition, we varied the number of nucleosome-level (at 20, 25 and 35, respectively) and domain-level (at 20, 25 and 35, respectively) states. The resulting states are also similar (Supplementary Figs 3 and 4).

After segmentation, consecutive identical states were stitched together, forming regions of variable size. Although the median size for a nucleosome-level state was ~600 bp (Supplementary Fig. 5a), a domain-level state may extend to over 100 kb regions, as is the case of the HOXB cluster (Fig. 1b,c). Importantly, these small- and large-scale structures were identified from a single model that decomposes the input signals into components of different spatial resolutions.

**Nucleosome-level states detect small-scale structure.** Using a similar strategy as in ChromHMM[7], we functionally annotated the nucleosome-level states, based on the combinatorial pattern of ChIP-seq signals (Fig. 2a), the spatial distribution (Supplementary Fig. 5c) as well as the enrichment of various functionally relevant elements (Fig. 2b). In the end, these 30 nucleosome-level states were annotated as 14 distinct functional categories (Fig. 2a). Specifically, states N1 and N2 were characterized by high intensity of H3K4me2 and H3K4me3, and therefore were annotated as

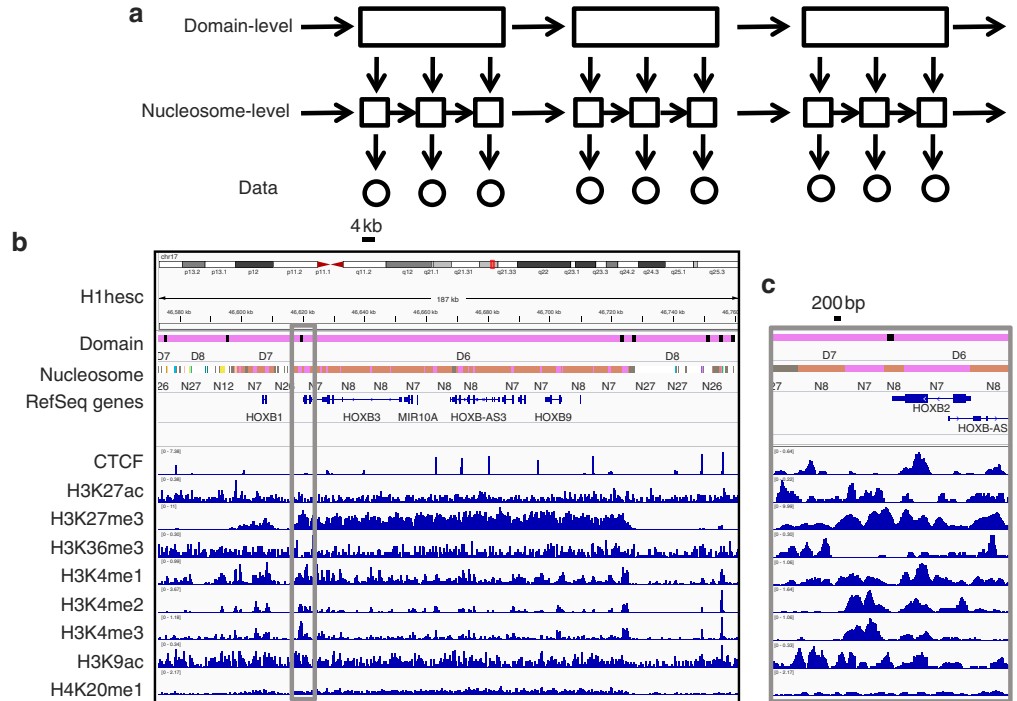

**Figure 1 | A schematic overview of diHMM. (a)** Shown is the underlying graphic model for diHMM with two levels of hidden states corresponding to the domain level (represented by rectangles) and nucleosome level (represented by squares), respectively. Multidimensional input ChIP-seq data are represented by circles. Arrows indicate the conditional dependence structure of diHMM. Nucleosome-level state transitions are dependent on the domain-level state at the end but not the initial position. The emission probability is conditionally independent of the domain-level state given the nucleosome-level state (see methods and Supplementary Fig. 1 for additional details). **(b)** Genome tracks displaying diHMM state calls in H1 cells for domain- and nucleosome-level states, and nine histone marks in the HOXB cluster region in chromosome 17. Grey box is expanded in **c** and shows a region of ∼8 kb. In the domain-level track black bars indicate transitions between different domains.

active promoters. Promoter flanking states (N3–N6) had predominantly H3K4me2, and were enriched around transcription start sites (TSSs) (Supplementary Fig. 5c). diHMM identified two nucleosome-level states (N7–N8) that were enriched in a repressive marker, H3K27me3, and an active marker, H3K4me2 or H3K4me1. Due to the spatial distribution difference, these states are annotated differently as bivalent promoters (N7) and poised enhancers (N8), respectively. Strong enhancer states (N9–N11) were associated with high H3K27ac and H3K4me1 signals, whereas weak enhancers (N12–N13) were enriched in H3K4me1. We found a category of transcribed enhancer states (N14–N19) that were enriched in gene body regions (Supplementary Fig. 5c), often associated with H3K36me3, H3K4me1 and sometimes in conjunction with H3K4me2. Transcriptional elongation states (N20–N21) were enriched in H3K36me3 but depleted in the enhancer markers. diHMM also found three states enriched in CTCF (N22–N24). Based on the spatial distributions, these states are further divided into two subcategories: CTCF promoter (N22) and CTCF (N23–N24) (Supplementary Fig. 5c). We also found a state (N25) that was enriched in only H4K20me1 and located downstream from TSS (Supplementary Fig. 5c). The polycomb repressed state (N26) was characterized by the enrichment of H3K27me3 and no other marks. The vast majority of the genome was characterized by a heterochromatin/low signal state (N27–N28). Finally, there were two infrequent states (N29–N30) characterized by the abundance of nearly all marks. These states typically fell in repetitive regions and therefore referred to as the repetitive/copy number variation (CNV) state.

Comparison of genomic coverage for nucleosome-level states in different cell types revealed some interesting features of chromatin organization (Fig. 2c). For instance, the bivalent promoter state was more prevalent in H1 cells, whereas strong enhancer and polycomb repressed states were more prevalent in GM12878 and K562 cells. Despite these notable differences, overall, nucleosome-level state usage was fairly similar between the different cell types considered in this study.

**Domain-level states detect large-scale structure.** Next, we annotated domain-level states based on their enrichment into different nucleosome-level states (Fig. 2d), transitions (Supplementary Fig. 6) and spatial distributions (Supplementary Fig. 5d). In total, we divided the domain-level states into 13 distinct functional categories. We found two kinds of domains enriched in nucleosome-level promoter states. One highly enriched in active promoter/promoter flanking states (N1–N5), and therefore called broad promoters domain (D1–D3); another one enriched in the flanking promoter state (N6) and with a significant overlap with exons, and therefore called promoters/exons domain (D4 and D5). Next, we identified two categories enriched in various repression-associated nucleosome-level states (bivalent promoter, poised enhancer, polycomb), and labelled them accordingly as bivalent promoter (D6–D8) and poised enhancer domains (D9), respectively. Attesting to the importance and complexity of enhancers in gene regulation, diHMM found nine domain-level states (D10–D18) enriched in enhancers that were further classified into three subcategories. super-enhancer domains (D10–D13) were highly enriched in strong enhancer (N9–N11), whereas upstream enhancer domains (D14 and D15) were enriched in weak enhancer (N12 and N13) and associated with being upstream from annotated TSS. A third enhancer domain category, which we called intron/enhancer (D16–D18), was mostly enriched in transcribed enhancer states (N14–N19)

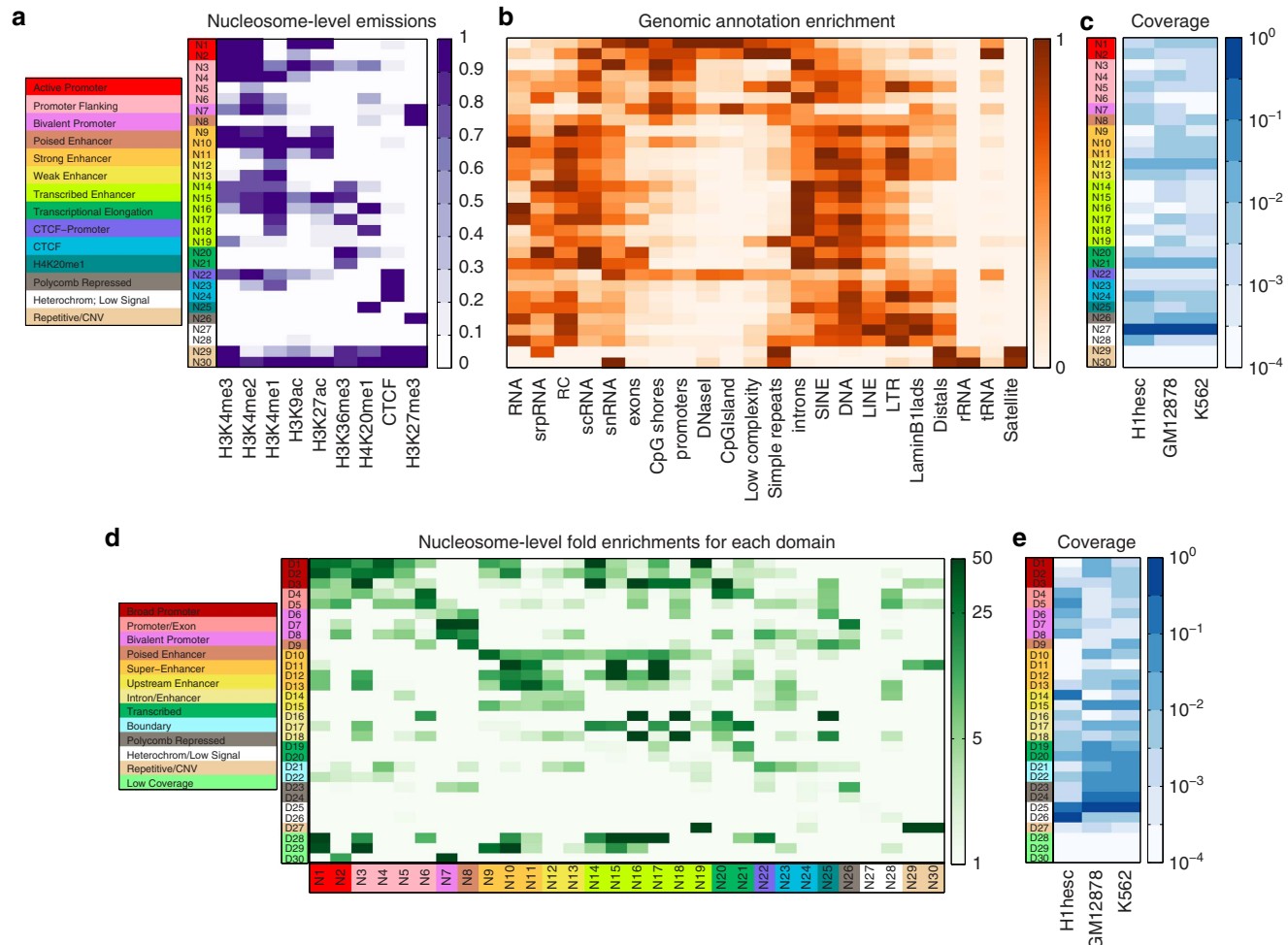

**Figure 2 | Annotation of the chromatin states identified by diHMM. (a)** Emission probability matrix for our diHMM model that contains 30 domain-level and 30 nucleosome-level states. The scale varies linearly between 0 (white) and 1 (dark purple). Colour legend on the left shows our nucleosome-level state annotations. **(b)** Genomic annotation enrichment for our 30 nucleosome-level states in all cell types combined. Each column shows relative enrichment in a linear scale between 0 (white) and 1 (dark orange). **(c)** Fraction of genomic coverage in each cell type for each nucleosome-level state. The scale varies logarithmically between $10^{-4}$ (white) and 1 (dark blue). **(d)** Significant fold enrichments for nucleosome- and domain-level combinations. Only combinations for which false discovery rate (FDR) < 0.01 (Fisher's exact test) are displayed above background level. The scale varies logarithmically between 1 (white) and 50 (dark green). Colour legend on the left shows our domain-level annotations. **(e)** Fraction of genomic coverage in each cell type for each domain-level state. The scale varies logarithmically between $10^{-4}$ (white) and 1 (dark blue).

and primarily located downstream from TSS. We found a transcribed domain (D19 and D20), which was enriched in the transcribed elongation state (N21) and distributed over a broad region downstream from TSS. The next category, which we called boundary domains, contained two domain-level states (D21 and D22) that were enriched in CTCF and located upstream from TSS. We found two polycomb repressed domains (D23 and D24) and two heterochromatin/low signal domains (D25 and D26) that were enriched in nucleosome-level polycomb and heterochromatin/low signal states, respectively. diHMM also captured regions enriched in satellite DNA and repetitive elements that were annotated as repetitive/CNV domains (D27). The last three domain-level states (D28–D30) were infrequent in the genome and assigned as low coverage states (Fig. 2e).

The overall usage of super-enhancer states (D10–D13) was much more prevalent in GM12878 and K562 cells compared with H1 (Fig. 2e) that agreed with previous observations[22]. Among these four states, only D13 was moderately enriched in H1 cells, whereas the other super-enhancer states were exclusively present in GM12878 and K562. Of note, D13 was distributed upstream from TSS, whereas the others were located in intronic regions

(Supplementary Fig. 5d), suggesting they may have different biological functions. Furthermore, poised enhancer and bivalent promoter states were more prevalent in H1. A subset of the corresponding loci, such as the HOXB gene cluster, switched to super-enhancer domains in differentiated cells (Supplementary Fig. 7a), and such transitions were associated with cell type-specific gene activation. In the meantime, polycomb repressed states were more prevalent in GM12878 and K562. Cell type-specific repression of these loci, such as BLK in K562 (Supplementary Fig. 7b) and the β-globin locus in GM12878 (Supplementary Fig. 7c), may play a role in suppressing gene expression program from alternative cell lineages. Altogether, these results show that our domains are able to capture functional differences among diverse regulatory elements in a cell type-specific manner.

**Context-dependent function of nucleosome-level states.** diHMM provides an opportunity to systematically investigate how the function of enhancer elements is influenced by the large-scale chromatin organization, an effect that cannot be evaluated

based on a single-scale model. For example, the enhancer state N13 was used in both poised enhancer (D8) and super-enhancer (D10) domains (Fig. 2d and Supplementary Fig. 6), but its spatial context was very different in these domains. In D8, it transitions to heterochromatin (N27, N28) and polycomb repressed state (N26), whereas in D10 it often transitions to strong enhancer states (N9–N11) or transcribed enhancer states (N14–N19). To test whether such contextual differences were functionally relevant, we divided the nucleosome-level enhancer states (N9–N13) into two broad categories, one associated with super-enhancer domains and the other with other domains, and compared the expression levels of their target genes. Remarkably, the gene expression levels corresponding to super-enhancer domain associated enhancers were much more cell-type specific (Fig. 3), indicating this subset of enhancers may play a more important role in maintenance of cell identity than other enhancers. This difference was not obvious for other enhancer-associated domains (poised enhancer, upstream enhancer and intron/enhancer) (Supplementary Fig. 8). We also compared our super-enhancer domains with the super-enhancers originally identified by the Lab of Young and colleagues[23] and found a high degree of overlap, hence justifying its name (Supplementary Figs 9a and 10). These domains also had a high degree of overlap with stretch enhancers[22] and broad H3K4me3 domains[24] (Supplementary Fig. 10). Next, we observed that downregulated genes were typically associated with bivalent promoter nucleosome-level states in the context of polycomb repressed domains (Fig. 3b). We repeated this analysis for other domain-level contexts and found a weaker trend for bivalent promoter domains (Fig. 3).

Although diHMM is not designed to predict long-range chromatin interactions, we expected certain relationships between diHMM domains and chromatin interaction patterns. A distinct feature in higher-order chromatin structure is that the compartmentalization into topologically associated domains (TADs), whose boundaries insulate chromatin interactions[13]. While diHMM domains are much smaller, we hypothesized that there may be distinct patterns associated with TAD boundaries that can be resolved at a 10 kb resolution. To test this hypothesis, we analysed a publicly available data set[15] containing high-resolution Hi-C data in two cell-types, GM12878 and K562, that are analyzed in this study. We found a strong bias of domain-level state transitions at TAD boundaries compared with the genomic background (for GM12878, fold change $= 1.9$; for K562, fold change $= 1.8$; in both cases $P$ value $< 2.2e - 16$, Fisher's Exact test) (Supplementary Fig. 11a). Similar bias were also found at chromatin loop anchors (for GM12878, fold change $= 1.6$; for K562, fold change $= 1.8$; in both cases, $P$ value $< 2.2e - 16$) (Supplementary Fig. 11b). We further analysed the association between domain-level states and chromatin interaction hubs, regions that are most enriched in chromatin interactions. Our previous analysis showed a significant association between chromatin interaction hubs and nucleosome-level enhancer elements[25]. Here we extended the analysis by comparing with the domain-level states. We found that the super-enhancer domains were moderate but statistically significantly (for

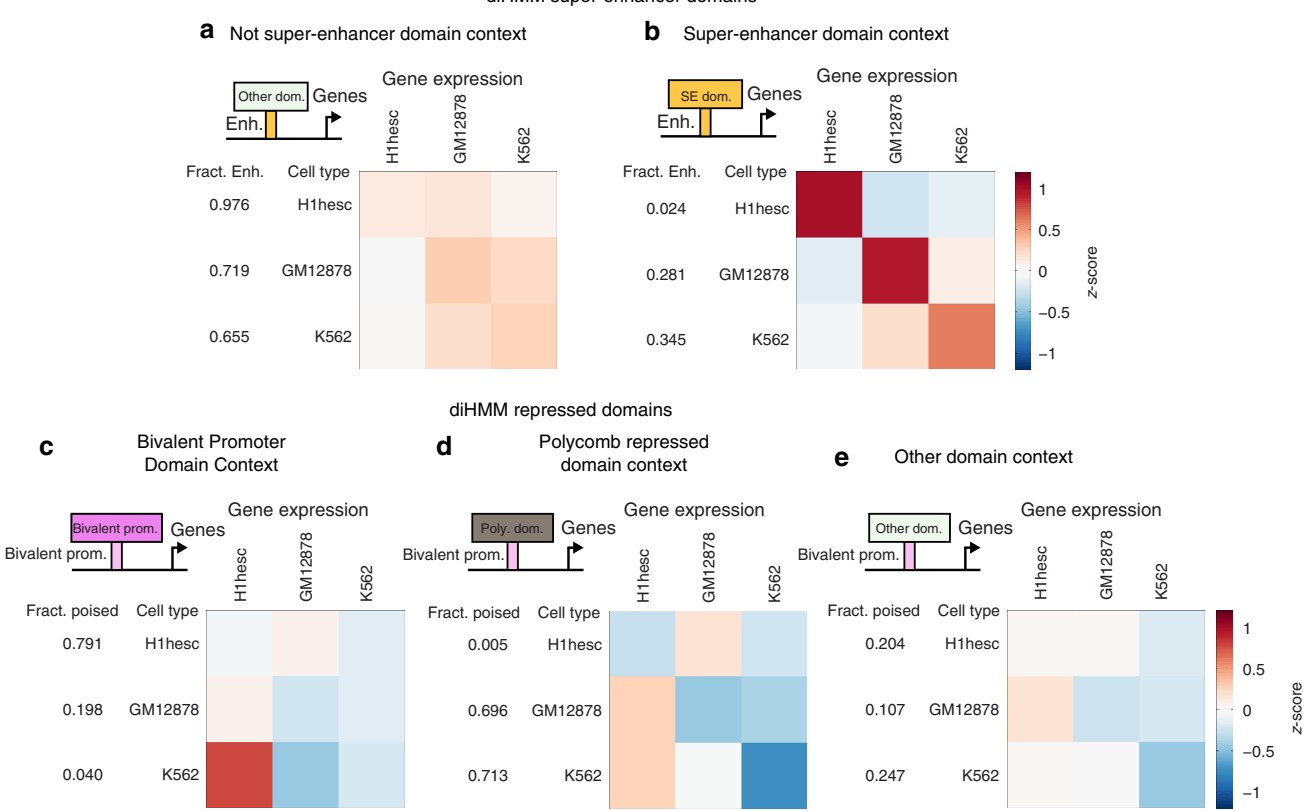

**Figure 3 | Context-specific functionality of diHMM nucleosome-level states.** (**a**,**b**) Heatmaps represent average gene expression ($z$-score for each gene and cell line obtained from a panel of 17 cell lines studied by ENCODE[2]) for genes mapped to enhancers in different domain contexts. In each row, genes are selected by proximity ($\pm 2$ kb from TSS) to nucleosome-level enhancers (states N9 to N13) in super-enhancer domains (D10–D13) or in the rest of the domains, as indicated by the small cartoon in each heatmap. Each column represents the average gene expression values for the different sets of genes when estimated in different cell lines. Numbers indicate the fraction of enhancers distributed between the different domains. (**c–e**) Heatmaps represent average gene expression for genes mapped to bivalent promoter state N7 in different domain contexts as indicated.

GM12878, fold change = 1.3; for K562, fold change = 1.2; in both cases, $P$ value < 2.2e − 16, Fisher's Exact test) enriched in hubs (Supplementary Fig. 11c). Overall, these results strongly indicate the regulatory potential of a genomic element is dependent not only on its associated marks but also on the broader spatial context.

**Comparison of diHMM with existing methods.** Existing chromatin-state annotation methods usually focus on a specific length scale. To see whether diHMM provides new insights, we selected a few representative methods and compared their results with diHMM. First, we compared the nucleosome-level annotations with chromHMM and Segway[10], two widely used methods for nucleosome-level chromatin-state annotations. We applied a 30-state ChromHMM to analyse the same data, and found that the nucleosome-level states agreed very well between diHMM and ChromHMM (Supplementary Fig. 12a,b). Segway is a dynamic Bayesian network-based chromatin-state segmentation method. It also has higher spatial resolution (at 10 bp) than chromHMM. We compared the chromatin-state annotations identified by diHMM and Segway. As expected, the agreement between the nucleosome-level chromatin states is significantly weaker, but the overall functional annotations are quite similar (Supplementary Fig. 13).

We wondered whether similar results regarding chromatin domains could be obtained by applying traditional models with different parameter settings. To this end, we adapted ChromHMM to identify domain-level states, using two alternative approaches: (1) We divided the genome into 4 kb bins, and applied a 30-state ChromHMM to segment the genome; and (2), we first applied ChromHMM to identify nucleosome-level states (with 200 bp resolution), stitched each set of 20 consecutive bins into a block, and applied $k$-centre to cluster the block-wide nucleosome-state patterns. We chose $k = 30$ so that the results were comparable.

We found significant discrepancies at the domain level between diHMM and the results for both (1) and (2) (Supplementary Fig. 12c,d). For both (1) and (2) the domain-level segmentations were more fragmented compared with diHMM (Supplementary Figs 5b and 14), and had lower enrichment in regulatory elements (Supplementary Fig. 8). In addition, although there was still significant bias of gene expression among different ChromHMM-derived domains in (1) and (2), the trend was much weaker compared with diHMM (Supplementary Fig. 15). Taken together, these results suggest the domain-level states identified by diHMM are more biologically meaningful.

Recently, a BCP model was developed to identify local domains (called BLOCKS) with similar histone modification patterns[20]. BCP is computationally less efficient than diHMM, and therefore we only trained a BCP model on 20 kb resolution signal on chromosome 17. This resulted in 25 BLOCKS with an average size of 3.2 Mb, which is about two orders of magnitude wider than diHMM. For comparison, we examined the diHMM domain-level state distribution near BLOCKS boundaries but were unable to find a significant association between the two methods, suggesting these two methods may identify complementary chromatin structures.

## Discussion

Cell-fate transitions are accompanied by extensive remodelling of chromatin architecture. While most studies have focused on nucleosome-scale dynamics, several experimental methods have revealed higher-order chromatin reorganization[26–28]. On the other hand, computational methods for chromatin-state annotation[4,5,29] analyse the data at a single length scale.

Therefore, diHMM fills an important methodological gap by providing a systematic modelling framework to simultaneously annotate chromatin states at multiple length scales. There are no minimum data requirement of diHMM. Indeed, it can even be applied to analyse a single mark. Here the domain-level states can be used to identify broad regions occupied by the mark (Supplementary Fig. 16). If a few marks are not measured for a cell type of interest, ChromImpute[30] can be used to impute the missing data before applying diHMM. Finally, while we have only focused on a two-level model implementation in this paper, it can be naturally extended to incorporate additional levels (see Methods for details).

The most extensively investigated chromatin state is the enhancer that plays an important role in cell type-specific gene regulation. At the nucleosome scale, enhancers are distinctly marked by H3K27ac and H3K4me1 (refs 9,31). At the domain level, our diHMM analysis has identified three distinct patterns of enhancer domains, super-enhancer, upstream enhancer and intron/enhancer, thereby unravelling significant complexity among different enhancers. We further find that the functionality of an enhancer strongly depends on the domain-level chromatin-state context, with the super-enhancer domain conferring the strongest regulatory potential. Our analysis is consistent with the recent discovery that multiple regulatory elements may cluster together, spanning over 10 kb regions, and cooperatively regulate cell identity[22,23,32]. Of note, the super-enhancer domain identified by diHMM differs from the traditional definition of super-enhancers, in that it describes a combinatorial pattern of multiple chromatin marks whereas the traditional definition is based on H3K27ac alone.

Long-range chromatin interactions play important roles in diverse biological processes including gene regulation, DNA replication and repair. Despite the rapid development of genomic technologies[14,33], it remains costly and challenging to profile genome-wide chromatin interactions at a high resolution. In the meantime, new computational methods have shown promise to predict chromatin interactions from ChIP-seq experiments[25]. The chromatin states identified by diHMM will provide useful features that will aid the development of new tools for predicting chromatin interactions, since the spatial resolution of the chromatin states at each level can be independently tuned to match the length scale of chromatin interactions.

Genome-wide association studies have shown that many of the disease-causing genetic variants are associated with noncoding regions[34]. While the function of the majority of these variants remains unknown, integration of genomic, epigenomic and transcriptomic data has strongly indicated that many play an important role in gene regulation[35]. It is important to recognize the intrinsic differences in temporal and spatial length scales among different data types. diHMM provides a coherent modelling framework to incorporate such differences.

## Methods

**Mathematical details of diHMM.** diHMM is a hierarchical hidden Markov model and can be used to incorporate multiple levels of hidden states. For simplicity, we only consider a two-level (nucleosome-level and domain-level) model in this paper (Fig. 1 and Supplementary Fig. 1), although the model can be generalized to include any number of layers as described in the following section. The ChIP-seq data were binarized in 200-base-pair bins with ChromHMM[21] using a Poisson background model and a threshold of $P$ value = $10^{-4}$, and the values at the $i$th bin are denoted by $x_i$, whereas the associated chromatin state is denoted by $\pi_i$ that contains two components, $j$ and $\mu$, corresponding to the nucleosome- and domain-level state, respectively. We use Latin indices for nucleosome-level states and Greek indices for domain-level states.

The basic assumptions in diHMM are similar to traditional HMMs[36]:

- Markov property $P(\pi_{i+1}|\pi_1, \ldots, \pi_i) = P(\pi_{i+1}|\pi_i)$
- Independence of observations $P(\mathbf{x}_i, \ldots, \mathbf{x}_L|\pi_1, \ldots, \pi_L) = \prod_{i=1}^{L} P(\mathbf{x}_i|\pi_i)$.

In addition, we also make the following specific assumptions about the relationship between different levels of hidden states:

- The emission probability, denoted by $e_k(\mathbf{b})$, is independent of the domain-level state, conditioned on the nucleosome-level state. That is,

$$e_k(\mathbf{b}) \equiv P(\mathbf{x}_i = \mathbf{b}|\pi_i = \{k, v\}) = P(\mathbf{x}_i = \mathbf{b}|\pi_i = \{k\}).$$

- Nucleosome-level transitions are domain dependent (indicated by $T_{N,jk}^v$, see later).
- Domain-level transitions can only occur at the end of blocks of size $D_S$, set to be 20 in this paper, that is, domain-level transitions can only occur every 20 bins. Since we use a bin size of 200 bp, this implies that the minimum domain size is 4 kb.

With these assumptions, transitions between states can be decomposed into nucleosome-level ($T_{N,jk}^v$) and domain-level ($T_{D,i}^{\mu v}$) transition matrices as follows:

- For positions for which $i$ is not a multiple of $D_S$, domain-level transitions are not possible, $T_{D,i}^{\mu v} = \delta^{\mu v}$, where $\delta^{\mu v}$ is the Kronecker delta and thus

$$P(\pi_{i+1} = \{k, v\}|\pi_i = \{j, v\}) = T_{N,jk}^v.$$

- For positions for which $i$ is a multiple of $D_s$, domain-level transitions are permitted, and thus

$$P(\pi_{i+1} = \{k, v\}|\pi_i = \{j, \mu\}) = T_{D,i}^{\mu v} T_{N,jk}^v.$$

Where we have taken the convention of using the nucleosome-level transitions corresponding to the final domain-level state $v$.
Finally, the initial state probabilities are

$$\pi_{1,j}^{\mu} = P(\pi_1 = \{j, \mu\}).$$

To train diHMM we extend standard dynamic programming techniques in HMMs[36], based on a combination of forward and backward algorithms. To avoid rounding errors it is important to scale the variables.

**Forward algorithm.** We define the forward variable for state $\{j, \mu\}$, at position $i$, on chromosome $n$ (of length $L$) as:

$$f_j^{n,\mu}(i) = P(\mathbf{x}_1^n, \mathbf{x}_2^n, \dots, \mathbf{x}_i^n, \pi_i^n = \{j, \mu\}).$$

The forward variable can be calculated recursively:
Initialization:

$$f_j^{n,\mu}(1) = \pi_{1,j}^{\mu} e_j(\mathbf{x}_1^n),$$

Induction ($i = 2, L$):

$$f_k^{n,v}(i) = e_k(\mathbf{x}_i^n) \sum_{j,\mu} \left[ f_j^{n,\mu}(i-1) T_{D,i}^{\mu v} T_{N,jk}^v \right],$$

Termination:

$$P(\mathbf{x}^n) = \sum_{j,\mu} f_j^{n,\mu}(L).$$

To avoid underflow errors we rescale the forward variables by using a series of scaling factors $s_i^n$, whose values will be determined later, so that the rescaled variables,

$$\hat{f}_j^{n,\mu}(i) = \frac{f_j^{n,\mu}(i)}{\prod_{k=1}^{i} s_k^n},$$

satisfy the following normalizing property,

$$\sum_{j,\mu} \hat{f}_j^{n,\mu}(i) = 1.$$

The induction formula for the rescaled variables becomes

$$\hat{f}_k^{n,v}(i) = \frac{1}{s_i^n} e_k(\mathbf{x}_i^n) \sum_{j,\mu} \left[ \hat{f}_j^{n,\mu}(i-1) T_{D,i}^{\mu v} T_{N,jk}^v \right].$$

Therefore, the values of $s_i^n$ can be solved as

$$s_i^n = \sum_{k,v} \left\{ e_k(\mathbf{x}_i^n) \sum_{j,\mu} \left[ \hat{f}_j^{n,\mu}(i-1) T_{D,i}^{\mu v} T_{N,jk}^v \right] \right\}.$$

The probability of the observed sequence can be calculated from the scaling variables as:

$$P(\mathbf{x}^n) = \sum_{j,\mu} f_j^{n,\mu}(L) = \sum_{j,\mu} \left[ \hat{f}_j^{n,\mu}(L) \prod_{k=1}^{L} s_k^n \right] = \prod_{k=1}^{L} s_k^n.$$

**Backward algorithm.** We define the backward variable for state $\{j, \mu\}$, at position $I$,

on chromosome $n$ (with $L$ bins) as:

$$b_j^{n,\mu}(i) = P(\mathbf{x}_{i+1}^n, \mathbf{x}_{i+2}^n, \dots, \mathbf{x}_L^n|\pi_i^n = \{j, \mu\}).$$

The backward variable can be calculated recursively:
Initialization:

$$b_j^{n,\mu}(L) = 1.$$

Induction ($i = L - 1, 1$):

$$b_j^{n,\mu}(i) = \sum_{k,v} \left[ T_{D,i}^{\mu v} T_{N,jk}^v b_k^{n,v}(i+1) e_k(\mathbf{x}_{i+1}^n) \right].$$

Termination:

$$P(\mathbf{x}^n) = \sum_{j,\mu} \left[ b_j^{n,\mu}(1) e_j(\mathbf{x}_1^n) \right].$$

As in the forward algorithm, it is beneficial to rescale the backward variables. In fact, using the scaling factors obtained from the forward algorithm,

$$\hat{b}_j^{n,\mu}(i) = \frac{b_j^{n,\mu}(i)}{\prod_{k=i+1}^{L} s_k^n}.$$

It can be shown that the following normalizing property holds:

$$\sum_{j,\mu} \hat{b}_j^{n,\mu}(i) = 1.$$

The induction formula for the rescaled backward variables is:

$$\hat{b}_j^{n,\mu}(i) = \frac{1}{s_{i+1}^n} \sum_{k,v} \left[ T_{D,i}^{\mu v} T_{N,jk}^v \hat{b}_k^{n,v}(i+1) e_k(\mathbf{x}_{i+1}^n) \right].$$

**Posterior probabilities.** We use the rescaled forward and backward variables to calculate the posterior probabilities

$$P(\pi_i^n = \{j, \mu\}|\mathbf{x}^n) = \frac{P(\pi_i^n = \{j, \mu\}, \mathbf{x}^n)}{P(x^n)} = \frac{f_j^{n,\mu}(i) b_j^{n,\mu}(i)}{\prod_{k=1}^{L} s_k^n}$$

$$= \frac{f_j^{n,\mu}(i)}{\prod_{k=1}^{i} s_k^n} \frac{b_j^{n,\mu}(i)}{\prod_{k=i+1}^{L} s_k^n} = \hat{f}_j^{n,\mu}(i) \hat{b}_j^{n,\mu}(i).$$

**Baum–Welch algorithm.** We train the model using the iterative Baum–Welch algorithm[36] with extension to incorporate the multilevel state structure. In this procedure, the training consists of a series of iterations in which the model parameters and state assignments are re-estimated sequentially, until convergence. In our model we start by using a state assignment obtained by clustering the bins at the 200 base-pair and 4 kb scales using the $k$-centre algorithm[37] and select the number of nucleosome and domains states. After the initial state assignment, the model parameters are re-estimated in the following way. At every iteration, we calculate the probabilities of finding two consecutive states

$$P(\pi_i^n = \{j, \mu\}, \pi_{i+1}^n = \{k, v\}|\mathbf{x}^n, \theta),$$

where $\theta$ represents all model parameters, by using the forward and backward variables as follows

$$\xi_i^n(\{j, \mu\}, \{k, v\}) = P(\pi_i^n = \{j, \mu\}, \pi_{i+1}^n = \{k, v\}|\mathbf{x}^n, \theta)$$

$$= \frac{P(\mathbf{x}^n, \pi_i^n = \{j, \mu\}, \pi_{i+1}^n = \{k, v\}|\theta)}{P(\mathbf{x}^n)}$$

$$= \frac{f_j^{n,\mu}(i) T_{D,i}^{\mu v} T_{N,jk}^v b_k^{n,v}(i+1) e_k(\mathbf{x}_{i+1}^n)}{P(\mathbf{x}^n)}$$

$$= \frac{\hat{f}_j^{n,\mu}(i) T_{D,i}^{\mu v} T_{N,jk}^v \hat{b}_k^{n,v}(i+1) e_k(\mathbf{x}_{i+1}^n)}{s_{i+1}^n}.$$

To update the domain-level transition probability $T_D^{\mu v}$, we sum over the marginal probabilities at the domain boundaries,

$$\chi_i^n(\{\mu\}, \{v\}) = P(\pi_i^n = \mu, \pi_{i+1}^n = v|\mathbf{x}^n, \theta)$$

$$= \sum_{j,k} P(\pi_i^n = \{j, \mu\}, \pi_{i+1}^n = \{k, v\}|\mathbf{x}^n, \theta)$$

$$= \sum_{j,k} \xi_i^n(\{j, \mu\}, \{k, v\}).$$

We have then

$$\bar{T}_D^{\mu v} = \frac{\sum_n \sum_{\{i|\text{mod}(i,D_S)=0\}} \chi_i^n(\{\mu\}, \{v\})}{\sum_\rho \sum_{n'} \sum_{\{j|\text{mod}(j,D_S)=0\}} \chi_j^{n'}(\{\mu\}, \{\rho\})}.$$

To update the nucleosome-level transition probability $T_{N,jk}^v$, we use a similar strategy, while marginalizing out $\mu$

$$\psi_i^n(\{j\}, \{k, v\}) = P(\pi_i^n = \{j\}, \pi_{i+1}^n = \{k, v\}|\mathbf{x}^n, \theta) = \sum_\mu \xi_i^n(\{j, \mu\}, \{k, v\}),$$

thus

$$\bar{T}^{v}_{N,jk} = \frac{\sum_n \sum_{i=1}^{L-1} \psi_i^n(\{j\},\{k,v\})}{\sum_l \sum_{n'} \sum_{j=1}^{L-1} \psi_j^{n'}(\{j\},\{l,v\})}.$$

To re-estimate the initial probabilities we average over the posterior probabilities at the first bin and for all chromosomes

$$\pi_{1,j}^{\mu} = \frac{1}{N} \sum_n \hat{f}_j^{n,\mu}(1)\hat{b}_j^{n,\mu}(1),$$

where $N$ is the total number of chromosomes.

The emission probabilities are updated by marginalizing out $\mu$ since in our model emissions only depend on the nucleosome-level state

$$E_k(\mathbf{b}) = \sum_n \sum_{\{i|\mathbf{x}_i^n = \mathbf{b}\}} \sum_{\mu} \hat{f}_j^{n,\mu}(i)\hat{b}_j^{n,\mu}(i),$$

giving

$$\bar{e}_k(\mathbf{b}) = \frac{E_k(\mathbf{b})}{\sum_{\mathbf{b}'} E_k(\mathbf{b}')}.$$

We apply the above procedure to analyse the combined ChIP-seq data set for H1hesc, GM12878 and K562, and obtain a single model that simultaneously annotates the chromatin states in these three cell lines. Due to computational constraints, we use chromosome 17 as the training data. It takes about 10 computer days to train the diHMM model on a computer with Linux CentOS release 6.6 (final), CPU Intel(R) Xeon(R) CPU X5650 @ 2.67 GHz, Mem 48G. The resulting model is applied to infer chromatin states in the whole genome that takes <2 h.

We test the robustness of diHMM by varying a number of parameters: (1) using chromosome 20 as the training data; (2) setting the number of nucleosome-level states at 20, 25 or 35; and (3) setting the number of domain-level states at 20, 25 or 35. The resulting chromatin-state assignment is compared with the original model (Supplementary Figs 3 and 4).

To quantify the degree of agreement between the chromatin-state annotations obtained from different models, or different parameter settings of the same model, we define a composite 'similarity score' that takes into account two complementary factors: (1) the similarity between the closest matching states and (2) the overall specificity of chromatin-state mapping. Mathematically, we represent the genome-wide distributions of each state $k$ as a numerical vector $X_k$, whose values are determined by the frequency of the state within each 4 kb window along the genome. To compare the annotations obtained from two models or settings, represented by $X$ and $Y$ respectively, we define the similarity score by using the following formula

$$\text{Similarity Score} = \frac{1}{K} \sum_{k=1}^{K} \left( \max_j \text{PCC}(X_k, Y_j) \text{Gini}(k, Y) \right)$$

where $\text{PCC}(X_k, Y_j)$ represents Pearson's correlation coefficient between the two vectors, and $\text{Gini}(k, Y)$ represents the Gini index of $Y$ conditioning on $X = k$.

**Generalization for incorporating additional levels of chromatin states.** In this paper, we focus on a two-level diHMM, but the modelling framework can be extended to incorporate any number of chromatin-state levels. Here we briefly outline the necessary steps for incorporating more than two levels. As in the two-level model, a higher-order chromatin state is assigned to each block of consecutive bins based on the combinatorial pattern of chromatin states at a lower level. The emission probability is solely determined by the chromatin states at the lowest level, whereas the state transition matrix is composed of multiple levels of transitions. We further assume that the interlevel coupling is restricted to neighbouring levels, that is, the nucleosome-level transition matrix is only dependent on the domain level, and so on. Model inference can be achieved in the same manner as described in the previous section—using the corresponding transition matrices. Of note, higher-level state transitions are only permitted at block boundaries.

**Data visualization.** To visualize genomic data and diHMM state calls we use Integrative Genomics Viewer[38,39]. To visualize nucleosome-level transitions for each domain we used circos[40].

**Functional enrichment analysis.** Enrichment of a particular functional label for a particular nucleosome- or domain-level state is calculated as $(m/n)/(M/N)$, where $m$ is the number of states overlapping the specific label, $n$ is the total number of 200 bp (for nucleosome-level enrichment) or 4 kb (for domains-level enrichment) bins of overlap, $M$ is the number of bins that the state occupies and $N$ is the total number of 200 bp (for nucleosome-level enrichment) or 4 kb (for domain-level enrichment) bins. Enrichment around TSS is calculated in a similar manner, but in this case based on the enrichment of the nucleosome- or domain-level states in the bins surrounding all RefSeq coding gene annotations. For visualization purposes all enrichments around TSS are normalized in a linear scale between 0 and 1.

**Gene expression analysis.** Microarray gene expression data in 19 human cell lines are obtained from ENCODE[2]. The gene expression values are converted into z-scores. Chromatin states are mapped to genes whose TSS are within ± 2 kb. For each state, the z-scores corresponding to all mapped genes are averaged.

**Relationships between diHMM domains and chromatin interaction patterns.** To compare the domain-level chromatin states with the three-dimensional chromatin structure, we analyse a public high-resolution Hi-C data set[15]. The chromatin interaction hubs are identified as described previously[25], Briefly, we first normalize the raw interaction matrix using the ICE (Iterative Correction and Eigenvector Decomposition) algorithm[41]. Then, we identify statistically significant chromatin interactions by using Fit-Hi-C[42]. We rank the 5 kb segments by the interaction frequency and define the top 10% as the hubs[25].

For hub enrichment analysis, all enhancers are divided into two non-overlapping groups: super-enhancer domains (diHMM domains D10–D13) and non-super-enhancer domains. The fold enrichment of hubs in enhancers in super-enhancer group over genome background (both groups) is defined as $(m/n)/(M/N)$, where $m$ and $M$ represent the number of enhancers that overlap with at least one hub in super-enhancer group and in both groups respectively, and $n$ and $N$ represent the number of enhancers in SE group and in both groups respectively.

**Data availability.** Aligned ChIP-seq reads for 9 chromatin marks (CTCF, H3K4me3, H3K4me2, H3K4me1, H3K9ac, H3K27ac, H3K36me3, H4K20me1 and H3K27me3) in H1, GM12878 and K562 cell lines are obtained from University of California at Santa Cruz ENCODE genome browser (http://genome.ucsc.edu/ENCODE)[2]. BAM files are first converted to BED files using bedtools[43], and all available replicates for each condition are subsequently merged. The microarray data for 19 cell lines (H1, HELA, HEPG2, HMEK, HUVEC, NHEK, CACO2, GM12878, GM06990, SKNSHRA, HRE, SAEC, BJ, K562, NHLF, H7, NHDFAd, NHA and HSMM) are also obtained from ENCODE at the same site. The intra-chromosomal raw interaction matrix in GM12878 and K562 at 5 kb resolution are downloaded from Gene Expression Omnibus with accession number GSE63525. The corresponding TAD and the chromatin loop locations are downloaded from the publication website[15]. The source code of diHMM is hosted at the following GitHub project: http://github.com/gcyuan/diHMM.

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

## Acknowledgements

We thank Dr Jessica Larson for helpful discussions. This work was supported by a Claudia Barr Award and NIH Grants R21HG006778 and R01HL119099 to G.-C.Y. K.G's research was supported by the NIH grant K25HL133599.

## Author contributions

E.M. and G.-C.Y. conceived and designed the project. E.M., W.M., J.H., K.G., L.P., J.W., M.K. and G.-C.Y. developed analytical methods. E.M., J.H., K.G., and L.P. wrote the analysis software. E.M. and J.H. analysed the data. E.M. and G.-C.Y. wrote the manuscript. M.K. and G.-C.Y. supervised the study. All authors edited the manuscript.

## Additional information

**Competing interests:** The authors declare no competing interests.

**Publisher's note**: 
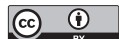

