## [Peer review file · Nature Communications]

Reviewers' comments:

Reviewer #1 (Remarks to the Author):

In the paper entitled "diHMM: Multi-scale chromatin state annotation using a hierarchical hidden Markov model" Marco and colleagues propose a computational data analysis framework (a two-level hidden Markov model, HMM) to process multiple genomic signals at two different scales. Overall, this is an important research problem, particularly now at post ENCODE time when various epigenetic data are becoming commonplace. Analysis of (epi)genomic data represents several challenges; here authors focus on the problem of analysing the epigenetic data that may have different length-scales. As authors note, very few methods have been proposed in literature that can handle epigenetic signals at multiple scales and thus "distinguish large domains from isolated elements of the same type".

The proposed method, diHMM, can be viewed as an extension of ChromHMM, a popular unsupervised method to annotate chromatin using multiple histone modification data sets. diHMM is defined to have two levels, a larger domain level (here 4-kb size) and a fine-grained nucleosome level (here 200bp size), such that larger domains can change state only at the end of the domain blocks and nucleosome level state transitions are domain-specific. Histone modification data is emitted from the nucleosome level. Similarly as in ChromHMM, histone modification data is assumed to be binarized (present/absent). Authors show how the standard model training algorithm for HMMs, the Baum-Welch algorithm, can be adapted to the diHMM model. Technical derivation seems correct (and is presented in an educative way). The current method is limited to two levels; thus it would be more appropriate to call the method a two-level chromatin state annotation method.

Authors use ENCODE data and apply diHMM to 9 histone modifications from three cell lines. Since the model now has two different length scales, or layers, the (hidden) domain level states (higher-level) can possibly detect higher order chromatin states while the (hidden) lower level can simultaneously model nucleosome level states. As expected, using the ENCODE data, authors show that their method can detect small-scale chromatin structures using the nucleosome-level states and, similarly, can detect large-scale chromatin structures using the domain level states. Authors compare their results to those obtained with the existing tool(s) at different resolutions of the input data; for nucleosome level the results agreed largely with the ChromHMM results; for domain level authors report "significant discrepancies" between diHMM and ChromHMM, the latter providing more fragmented domain-level segmentations and showing less enrichment in regulatory elements (taken from a previous study).

The most interesting results are presented in "Context-Dependent function of nucleosome-level states" section. For example, authors show that one of the nucleosome-level enhancer states (N13; enrichment of H3K4me1, no H3K27ac) was included in two different domain-level states (D8 and D10). On the other hand, as authors discussed in manuscript, this is perhaps not so surprising because active enhancers (marked with H3K27ac) can be flanked by other enhancer associated marks (e.g. H3K4me1); or conversely, enhancer mark H3K4me1 may or may not be flanked by active enhancer mark depending on the activity of the regulatory region, for example. Nevertheless, authors then further showed that N13 enhancer states in D10 domain (so called super-enhancer domain) were associated with

higher gene expression of nearby genes compared with N13 enhancers in other domains. (It would perhaps be more compelling to show this comparison between "super enhancer domain" and "other enhancer domain(s)" instead of "super enhancer domain" vs. "all other domain(s)".) A similar behaviour (except for gene expression down-regulation) was also observed for nucleosome level state "bivalent promoters". Another association results was shown between enhancer domains and 3-D chromatin interaction hubs, although the signal appears very weak since super enhancer states in H1 cell line has a higher enrichment of hubs (than K562) even though the 3-D interaction hubs were quantified in the two other cell lines.

Authors work on developing a two level HMM model for histone modification data is appreciated; however, the current manuscript version falls short in convincingly demonstrating that their diHMM method truly provides multi-scale information about the chromatin state. In particular, it is unclear that what is the significance of the reported results and associations.

Reviewer #2 (Remarks to the Author):

Summary:

Marco et al. describe diHMM, a novel segmentation approach that identifies chromatin states at the level of nucleosomes and broad domains simultaneously. For this purpose they employ a hierarchical Hidden Markov Model in which transitions between nucleosome-level states are governed by domain-level states. They apply diHMM to the three tier 1 ENCODE cell lines, provide a characterization of nucleosome and domain-level chromatin states and compare their method to alternative parametrizations of the chromHMM model.

This novel approach identifies links between nucleosome-level chromatin and broad domains. The methods appear statistically sound and are well-explained. However, the study could be improved in terms of evaluation of robustness, alternative parametrization, software usability and characterization of chromatin states in the context of cell-specific chromatin interactions.

General Comments:

- The parametrization of the model is not motivated in the manuscript. I.e. what is the rationale for choosing 30 nucleosome, 30 domain states and 20 nucleosome bins per domain bin? What other parametrizations were evaluated?
- The authors mention that multiple layers of hidden states can be modeled. Where different numbers of levels evaluated?
- Why was chromosome 17 selected for model training?
- Many model parameters have to be fitted (a separate nucleosome transition matrix for each domain state). How does the method address the issue of over-training?
- The manuscript would benefit from a benchmarking of computational performance. What are the time and resource requirements for model training and applying the method to an entire genome?
- Software usability: I was not able to evaluate the software itself, since I did not have access to a Matlab license. A stand-alone software-tool could be considered.
- The last commit to the git repository was over one year ago. This raises doubts on the mid to long-term usability and maintenance of the software.
- How are the three cells different in their chromatin states? The authors describe differences in the global prevalence of states. How would the authors interpret these changes biologically? What about changes in specific loci?
- On the nucleosome level, chromHMM is conceptually similar to the diHMM. How does the diHMM compare to other segmentation approaches? E.g. methods based on numeric rather than binarized data, e.g. Segway (doi:10.1038/nmeth.1937) or EpiCSeq (doi:10.1186/s13059-015-0708-z)?
- Is there a relation of domain states from diHMM to domains identified from chromatin interaction data (HiC, etc.)? These data are available from ENCODE and it would be interesting to compare them.

Specific Comments:

- The state names are somewhat misleading. For instance, the states that the authors term "promoters" (eg. bivalent promoters, CTCF promoters) are only overlapping with promoters to a certain degree.

- page 2: "At the basic level, chromatin structure is described by a one-dimensional series of nucleosome positions": this sentence could be misinterpreted. The series is not truly one-dimensional, but multi-dimensional along the one-dimensional genome coordinate.
- fig 1: legend (b): the term "histone mark" should be avoided, since it wrongly includes CTCF here
- page 5: fig 1e (referred to in the text) does not exist. The authors probably mean fig 2e
- fig 3: The description is somewhat unclear. Do the enhancers and repressed states (rows in the heatmaps) correspond to the states specific to each cell type?
- fig S4 is barely readable

Reviewer #3 (Remarks to the Author):

The manuscript titled “diHMM: Multi-scale chromatin state annotation using a hierarchical hidden Markov model” by Dr Yuan and co-workers present an approach to infer chromatin states using histone marks. The manuscript is well written and the results suggest the method proposed may help in the determination of the higher-order features for chromatin state annotation.

Major concerns:

The main idea of diHMM is to capture chromatin states at two levels: 1. nucleosome-level information, which can be done with ChromHMM.

2. domain-level information, which can be experimentally demonstrated with Hi-C and computationally predicted by BLOCKS using BCP model. It would be great if the performance of diHMM is super on both levels, or at least on one.

1. The author compared the overlaps between diHMM results and Hi-C interactions (Fig. S7). The enrichment of hubs for nucleosome-level enhancer states in diHMM in super-enhancer domains is in the range of 1.09 to 1.46. The numbers are quite low, and no statistical tests were provided.

2. diHMM predicts chromatin states at domain-level. The authors may want to make it clear whether diHMM can predict the interactions between domains and thus provide similar information as Hi-C for high-order interactions. If not, the authors may want to revise the line 45-47 in the introduction section.

3. Annotation of the chromatin states provided by diHMM relies on the prior knowledge about histone marks. The authors provided 30 nucleosome-level and 30 domain-level states. It would be very useful if authors can provide a clear guide to understand the biological functions of these 30 states. i.e. at nucleosome-level, N1 and N2 were used to represent “Active promoter”. It seems that some active promoters lack of H3K27ac modifications.

4. What is the minimum data requirement on histone modifications for diHMM? The authors used 9 chromatin marks. If diHMM is very sensitive to missing data, it will greatly limit the range of application.

Minor issues:

1. A comparison between diHMM and BLOCKS would be desired.

2. To determine domain-level states, 20 consecutive 200 bp bins were used in the study. In Fig S8, the ranges of domain length vary significantly. A dynamic sliding window instead of 20 bins may provide a better algorithm.

3. In Fig 3, the authors divided nucleosome-level enhancers into two categories: super-enhancer vs all others. It is not a surprise to see the striking differences between the two

categories. It would be more reasonable to divide “strong enhancer” at nucleosome -level into two groups: super-enhancer vs not-super-enhancer for comparison.

Reviewer #4 (Remarks to the Author):

Marco, Meuleman, Huang, Glass, Pinello, Wang, Kellis, and Yuan describe a new method for multiscale chromatin segmentation based on multivariate epigenomic profiling data, and the application of the method to analyze epigenomic data from three ENCODE human cell lines. They compared the results that they obtained from applying the new segmentation method with results that they obtained using a previously described single-scale chromatin segmentation method, ChromHMM. In the comparison, they used both gene expression data as well as predicted super-enhancer element locations (from an earlier study, referenced as [27] in this article, that was based on local enhancer density and H3K27ac ChIP-seq signal) for independent validation. The new chromatin segmentation method, diHMM, is based on a hierarchical hidden Markov model, with maximum-likelihood used for parameter estimation. The diHMM implementation used in this work incorporates both a nucleosome-level (200 bp bin size) layer and a domain-level (4 kbp bin size) layer, though the general approach is not restricted to those specific bin sizes or layer types. The authors' key findings are that the use of diHMM (i) enables the detection of associations between specific nucleosome-level states and domain-level states, (ii) enables the discrimination, based on the local domain state (i.e., whether or not the local domain state is "super-enhancer"), of enhancer states that are more likely to be associated with higher levels of transcript abundance for the nearest gene; (iii) produces domain-level chromatin segmentations that are less fragmented and more biologically meaningful than those that are mapped using a single-scale HMM.

The work is highly original, timely, and would be of strong interest to a broad scientific readership in genomics, computational biology, and cell biology. The methodology and datasets used are, overall, appropriate to the problem. The overall statistical approach is judged to be sound, although two suggested statistical tests are included under 'minor issues/comments'. Overall, the conclusions of the article are well-supported by the described studies (only a few suggested additional pieces of information are listed under 'minor issues/comment'). The article appropriately cites previous work in the field. In terms of clarity, the article could use some minor improvements (specifically detailed under 'minor issues/comments') but there are no major issues with clarity, in this reviewer's opinion.

Minor issues/comments:

1. lines 55-56: "flexible modeling framework that systematically identifies combinatorial patterns at multiple length scales"-- it would be helpful to be a bit more precise here about the specific way in which diHMM improves over current popular approaches (vs. just enumerating positive attributes of diHMM). Are the authors claiming that diHMM is more flexible than previous approaches? More systematic? (evidence or an argument to substantiate that claim would be helpful in that case). It seems to this reviewer that the primary and most interesting way in which diHMM extends beyond previous approaches is that it gives a principled framework for detecting latent domain states and their associations with specific nucleosome-scale chromatin states.

2. line 77: Authors are commended for releasing the source code on GitHub. While this reviewer recognizes that research code is different from commercial software, readers may benefit from a bit of improvement to the code header comments and a bit of code clean-up

(e.g., to remove "code introns" that have been commented out, or otherwise to explain their importance).

3. line 82: Would be nice for authors to briefly share their thoughts/reasoning on the origin of the choice of "30 states" for nucleosomes and domains, especially for a non-expert reader.

4. line 91, x-ref to Fig. 2a: are the data in Fig. 2a for all three cell lines, averaged? Clarification would be helpful, perhaps in the caption.

5. line 140, figure cross-reference "Fig. 1e": perhaps the authors meant "Fig. 2e" here?

6. line 185, figure cross-reference "Fig. S8"; perhaps the authors meant "Fig. S2b and S8" here? Seems necessary to reference Fig. S2b as well as S8 in order to see the fragmentation.

7. line 185 (Fig. S5 cross-ref): would be helpful to quantify the enrichment with some kind of overall test statistic.

8. line 187 (Fig. S9 cross-ref): would be helpful to quantify the overall effect where it says "the trend was much weaker compared to diHMM", using some kind of test statistic

9. line 233, "index i": took this reviewer a bit of forward-reading to grasp that "i" refers to the 200 bp bin; would be good to clarify this.

10. line 243, " $e_k(b)$ ": maybe state that this is the emission probability? Then it is clear why the symbol "e" is being used. Also "depend only on nucleosome-level state" is a bit vague. I think what is meant is that the observations are independent of the domain state, conditioned on the nucleosome-level state.

11. line 269: it would be helpful to introduce the scaling variable symbol s_i^n before it is used in the formula; the "meaning" of the symbol doesn't become apparent until line 272

12. line 314: a bit of explanation as to why "chromosome 17" was selected would be helpful, as well as some comment on whether the maximum-likelihood model based on the chromosome 17 epigenomic data would be expected to approximately maximize the likelihood for the data on another chromosome. As a related question, how variable are the principal results if training data from a different chromosome were used?

13. line 335: what if more than one gene is within +/- 2kb of the bin? Also, please clarify if log₂-transformed expression levels were converted to z-score (as would seem sensible), vs. non-log (i.e., "linear scale") expression levels.

14. lines 179-182: a bit more detailed explanation of precisely how (i) and (ii) differ, and what hypothesis is being tested by comparing them, would really help. So in (i), the k-center clustering is done on the ChIP-seq state information within a 4 kb block, and in (ii), the k-center clustering is done on the nucleosome states for the bins within the 4 kb block?

15. Some overall quantification of how many 4 kb bins "switch color" between ChromHMM-

4k and diHMM would be interesting to see; it is hard to guess from the heatmaps in Fig. S3c-d.

Response to Review #1

Remarks to the Author:

In the paper entitled "diHMM: Multi-scale chromatin state annotation using a hierarchical hidden Markov model" Marco and colleagues propose a computational data analysis framework (a two-level hidden Markov model, HMM) to process multiple genomic signals at two different scales. Overall, this is an important research problem, particularly now at post ENCODE time when various epigenetic data are becoming commonplace. Analysis of (epi)genomic data represents several challenges; here authors focus on the problem of analysing the epigenetic data that may have different length-scales. As authors note, very few methods have been proposed in literature that can handle epigenetic signals at multiple scales and thus "distinguish large domains from isolated elements of the same type".

The proposed method, diHMM, can be viewed as an extension of ChromHMM, a popular unsupervised method to annotate chromatin using multiple histone modification data sets. diHMM is defined to have two levels, a larger domain level (here 4kb size) and a fine-grained nucleosome level (here 200bp size), such that larger domains can change state only at the end of the domain blocks and nucleosome level state transitions are domain-specific. Histone modification data is emitted from the nucleosome level. Similarly as in ChromHMM, histone modification data is assumed to be binarized (present/absent). Authors show how the standard model training algorithm for HMMs, the Baum-Welch algorithm, can be adapted to the diHMM model. Technical derivation seems correct (and is presented in an educative way). The current method is limited to two levels; thus it would be more appropriate to call the method a two-level chromatin state annotation method.

Response: We thank the reviewer for this important question. While our data analysis focuses on a two-level implementation due to its simplicity and interpretability, we would also like to point out that the modeling framework can incorporate any number of layers. To make this point more obvious, we have added an additional Method section, entitled "Generalization for Incorporating Additional Levels of Chromatin States", to describe the mathematical details. The most notable modification is to select an appropriate form of the transition matrix in order to take into account state transitions at all layers in the model. The flexibility of incorporating any chromatin state levels is a major strength of diHMM.

Authors use ENCODE data and apply diHMM to 9 histone modifications from three cell lines. Since the model now has two different length scales, or layers, the (hidden) domain level states (higher-level) can possibly detect higher order chromatin states while the (hidden) lower level can simultaneously model nucleosome level states. As expected, using the ENCODE data, authors show that their method can detect small-scale chromatin structures using the nucleosome-level states and, similarly, can detect large-scale chromatin structures using the domain level states. Authors compare their results to those obtained with the existing tool(s) at different resolutions of the input data; for nucleosome level the results agreed largely with the

ChromHMM results; for domain level authors report "significant discrepancies" between diHMM and ChromHMM, the latter providing more fragmented domain-level segmentations and showing less enrichment in regulatory elements (taken from a previous study).

Response: Yes, this is an accurate interpretation of our analysis and results.

The most interesting results are presented in "Context-Dependent function of nucleosome-level states" section. For example, authors show that one of the nucleosome-level enhancer states (N13; enrichment of H3K4me1, no H3K27ac) was included in two different domain-level states (D8 and D10). On the other hand, as authors discussed in manuscript, this is perhaps not so surprising because active enhancers (marked with H3K27ac) can be flanked by other enhancer associated marks (e.g. H3K4me1); or conversely, enhancer mark H3K4me1 may or may not be flanked by active enhancer mark depending on the activity of the regulatory region, for example. Nevertheless, authors then further showed that N13 enhancer states in D10 domain (so called super-enhancer domain) were associated with higher gene expression of nearby genes compared with N13 enhancers in other domains. (It would perhaps be more compelling to show this comparison between "super enhancer domain" and "other enhancer domain(s)" instead of "super enhancer domain" vs. "all other domain(s)".) A similar behaviour (except for gene expression down-regulation) was also observed for nucleosome level state "bivalent promoters". Another association results was shown between enhancer domains and 3-D chromatin interaction hubs, although the signal appears very weak since super enhancer states in H1 cell line has a higher enrichment of hubs (than K562) even though the 3-D interaction hubs were quantified in the two other cell lines.

Response: As suggested, we have refined our analysis by comparing the enhancers in SE domains with three other enhancer-associated domains, including Poised Enhancer, Upstream Enhancer, and Intron/Enhancer. As shown in Figure S8, the association with cell-type specific gene expression is much weaker compared to SE enhancers.

As the reviewer pointed out, the association between SE and chromatin interaction hubs is somewhat weak, although these associations are statistically significant (p -value $< 2.2e-16$, Fisher exact test). This is not surprising because SE domains are typically broad (~10 kb) and contain diverse nucleosome-level elements, whereas a large number of chromatin interaction hubs are not associated with enhancers (ref. 26). The signal is further attenuated by the relatively large number of SE domains identified in GM12878 and K562 cells.

Authors work on developing a two level HMM model for histone modification data is appreciated; however, the current manuscript version falls short in convincingly demonstrating that their diHMM method truly provides multi-scale information about the chromatin state. In particular, it is unclear that what is the significance of the reported results and associations.

Response: We appreciate the reviewer's sentiment and agree that our analysis has not fully addressed the role of multi-scale chromatin state information in gene regulation. Our analysis is limited by the numerical inefficiency of the current implementation of diHMM. As a result, we have only analyzed and compared a small number of cell lines. We plan to significantly improve the numerical efficiency of diHMM by using a more efficient programming language in future work. If we can achieve this goal, we would then apply diHMM to the rich datasets generated by ENCODE and Epigenome Roadmap Project. This would likely provide new insights the role of domain-level chromatin states in regulating cell-type specific gene expression programs.

Nonetheless, our work presented here is significant in that it provides a flexible, generally applicable, modeling framework for systematic annotation of multi-scale chromatin states. Our integrative analyses of ChIP-seq, gene expression, and Hi-C data validate the functional relevance of the detected chromatin states. Importantly, we provide evidence that diHMM identifies context-dependent function of nucleosome-level states, which cannot be achieved by using any method based on a single length-scale.

Response to Review #2

Remarks to the Author:

Summary:

Marco et al. describe diHMM, a novel segmentation approach that identifies chromatin states at the level of nucleosomes and broad domains simultaneously. For this purpose they employ a hierarchical Hidden Markov Model in which transitions between nucleosome-level states are governed by domain-level states. They apply diHMM to the three tier 1 ENCODE cell lines, provide a characterization of nucleosome and domain-level chromatin states and compare their method to alternative parametrizations of the chromHMM model.

This novel approach identifies links between nucleosome-level chromatin and broad domains. The methods appear statistically sound and are well-explained. However, the study could be improved in terms of evaluation of robustness, alternative parametrization, software usability and characterization of chromatin states in the context of cell-specific chromatin interactions.

Response: We appreciate the reviewer's positive evaluation of our method as well as the helpful suggestions. As discussed below, we have carried out a number of additional analyses to systematically evaluate the model robustness and parametrization. We also refined the functional characterization by dissecting the differences between the enhancer elements in SE and other enhancer domains. The details are described in our response to the more specific comments below. Taken together, these additional analyses have provided stronger support to the utility of diHMM. As the reviewer pointed out, our method still has a number of limitations, including numerical efficiency and accessibility. These issues are more complex and will be addressed in future work.

General Comments:

- The parametrization of the model is not motivated in the manuscript. I.e. what is the rationale for choosing 30 nucleosome, 30 domain states and 20 nucleosome bins per domain bin? What other parametrizations were evaluated?

Response: The reviewer has pointed out a challenging problem in chromatin state segmentation. We are not aware of any effective method that can determine the number of chromatin states with sufficient mathematical rigor as well as easy biological interpretability. In previous work (Ernst & Kellis, Nature Biotech, 2010), we and other groups have used a number of statistical criteria to determine the optimal number of chromatin states, such as Bayesian Information Criterion, posterior probabilities, and classification consistency. However, we found that these approaches tend to result in a model that is too complex to be easily associated with biological functions. A likely reason is that these statistical tests often make idealized assumptions about the ChIP-seq data which are often violated in reality. Following previous studies (doi:10.1038/nmeth.1906, doi: 10.1038/nature09725. doi: 10.1038/nmeth.1937), we used an intuitive approach in order to capture the main signals by using a small number of chromatin states. Our previous experience (doi:10.1038/nmeth.1906, doi: 10.1038/nature09725.) suggests that a reasonable range is between 10 to 50 chromatin states for 9 marks. For simplicity, we picked 30 states at both nucleosome and domain levels. This choice is justified by comparing with known combinatorial histone modification patterns, spatial location with respect to transcription start and end sites, as well as enrichment of known genomic elements (Figs. 2 and S5).

In order to test the robustness of model parameterization, we repeated our analyses by using multiple model configurations, varying the number of chromatin states at both nucleosome- and domain-levels. In all cases, we found that the results are highly reproducible (Figures S3,S4) . Furthermore, we trained our model based on a different chromosome, chr 20, and found good agreement with the original segmentation (Fig. S2). Taken together, these results strongly support the robustness of diHMM.

- The authors mention that multiple layers of hidden states can be modeled. Where different numbers of levels evaluated?

Response: While we focused on a two-layer model to analyze the ChIP-seq data, the model can be easily extended to incorporate additional layers. To make this point more obvious, we have added a Method subsection, which is entitled “Generalization for Incorporating Additional Levels of Chromatin States”, in the revised manuscript to describe the mathematical details. In general, the exact model setting, including the number of levels, should be determined based on the range of length scales of interest and the biological questions at hand.

- *Why was chromosome 17 selected for model training?*

Response: We chose chromosome 17 for model training due to numerical efficiency consideration. Ideally we would like to train our model genome-wide. However, this was infeasible because the current implementation of diHMM was slow. Therefore, we arbitrarily selected a relatively short chromosome to train our model. It took over 10 computer days to complete modeling training and about 1 to 2 additional hours to annotate chromatin states genome-wide. To test for robustness with respect to the training data, we created another model using chromosome 20 as the training set. The chromatin states obtained from these independent training datasets are quite similar (Fig. S2).

- *Many model parameters have to be fitted (a separate nucleosome transition matrix for each domain state). How does the method address the issue of over-training?*

Response: We thank the reviewer for his/her attention to rigor. Our model indeed contains many parameters; therefore over-fitting is a valid concern. However, >90% of the parameters have values close to zero, suggesting they may not contribute significantly to the model outcome. To directly evaluate the risk of over-fitting, we compared the chromatin states obtained from two independent training datasets, corresponding to chr 17 and chr 20, respectively. We find that the results are highly reproducible (Fig. S2). These analyses suggest over-fitting may not be a significant issue.

- *The manuscript would benefit from a benchmarking of computational performance. What are the time and resource requirements for model training and applying the method to an entire genome?*

Response: We benchmarked the computational performance of our model by using a computer with Linux CentOS release 6.6 (final), CPU Intel(R) Xeon(R) CPU X5650 @ 2.67GHz, Mem 48G . It took about 10 days to train the diHMM model based on chr17 data, while using the trained model to segment the chromatin states genome-wide took less than two hours. These benchmarks are included in the revised manuscript (Methods section).

We are exploring different ways to improve numerical efficiency, such as using high-performance MATLAB and rewrite the code using a more efficient computer language. If successful, we will update the diHMM Github website in a timely manner.

- *Software usability: I was not able to evaluate the software itself, since I did not have access to a Matlab license. A stand-alone software-tool could be considered.*

Response: We thank the reviewer for evaluating diHMM and apologize for the difficulty he/she has experienced. We implemented diHMM in MATLAB because it is a widely-used and user-friendly computing environment by physical scientists and engineers. However, the fact that

MATLAB is not freely available limits its accessibility especially to the developing world. We thank the reviewer for pointing out this important limitation and plan to address this issue in the future.

- The last commit to the git repository was over one year ago. This raises doubts on the mid to long-term usability and maintenance of the software.

Response: We appreciate the reviewer's concern regarding long-term software maintenance. The co-senior authors (Kellis and Yuan) are committed to long-term usability and maintenance of any software developed by our groups and we both have strong track records.

While git commit is an obvious metric for maintenance effort, it does not reflect many other effort we have spent to improve diHMM. During the past year, we have been focusing on dissecting the biological functions of the diHMM chromatin states, integrating various data-types in the public domain. These analyses are important for us (and potential users) to interpret the biological meaning of these states and their role in gene regulation. In the meantime, we have been monitoring any potential issues from external users and helped a few users with installation and analysis related issues. As the reviewer may have noticed, there are no unsolved issues at the github site. As a significant extension of chromHMM, which has been widely used by the community, long-term maintenance of diHMM is a high priority.

- How are the three cells different in their chromatin states? The authors describe differences in the global prevalence of states. How would the authors interpret these changes biologically? What about changes in specific loci?

Response: Fig. 2e indicates that the global distribution of chromatin states vary quite significantly between the three cell lines. Fig. S7 further shows the variation at a few well-characterized loci playing important role in the maintenance of cell identity. At these loci, the chromatin states change from an inactive to an active chromatin state, which is consistent with cell-type specific gene expression level changes.

- On the nucleosome level, chromHMM is conceptually similar to the diHMM. How does the diHMM compare to other segmentation approaches? E.g. methods based on numeric rather than binarized data, e.g. Segway (doi:10.1038/nmeth.1937) or EpiCSeq (doi:10.1186/s13059-015-0708-z)?

Response: We have compared with Segway as suggested. As expected, there is an overall agreement between diHMM and Segway (Fig. S13), especially after merging the states from the same functional categories. However, the agreement is not as obvious as in the diHMM vs ChromHMM comparison, as expected. We also tried to compare diHMM with EpiCSeq but failed due to installation error.

- Is there a relation of domain states from diHMM to domains identified from chromatin interaction data (HiC, etc.)? These data are available from ENCODE and it would be interesting to compare them.

Response: We thank the reviewer for his/her interesting suggestion and examined the relationship between diHMM and the Hi-C derived topological associated domains (TAD). The average size of a TAD domain is 250 kb, whereas an average diHMM is 25 kb. Therefore, TAD domains represent one order of magnitude higher-order chromatin structures as compared to diHMM domains. We examined the diHMM state distribution near TAD boundaries, and found an elevated probability for diHMM domain state switches at the TAD boundaries (Fig. S11). In addition, we found association between SE domains and chromatin interaction hubs (Fig. S11).

Specific Comments:

- The state names are somewhat misleading. For instance, the states that the authors term "promoters" (eg. bivalent promoters, CTCF promoters) are only overlapping with promoters to a certain degree.

Response: We agree with the reviewer the name assignment is not perfect. diHMM is an extension of chromHMM, therefore we chose the naming system to be consistent chromHMM (such as Ernst et al, Nature, 2011). Using a different system might introduce confusion for users who are already using chromHMM. On the other hand, we welcome alternative suggestions for improving the naming system.

- page 2: "At the basic level, chromatin structure is described by a one-dimensional series of nucleosome positions": this sentence could be misinterpreted. The series is not truly one-dimensional, but multi-dimensional along the one-dimensional genome coordinate.

Response: We thank the reviewer for the suggestion and have changed the text as suggested.

- fig 1: legend (b): the term "histone mark" should be avoided, since it wrongly includes CTCF here

Response: We replaced the term "histone mark" to "mark" to avoid potential confusion.

- page 5: fig 1e (referred to in the text) does not exist. The authors probably mean fig 2e

Response: Yes. We have corrected this error in this revision.

- fig 3: The description is somewhat unclear. Do the enhancers and repressed states (rows in the heatmaps) correspond to the states specific to each cell type?

Response: Yes, the enhancers and repressed states selected in each row correspond to the total number of states in each cell type, including both common and specific states. We have revised the Figure legend to clarify.

- fig S4 is barely readable

Response: We apologize for the small font size for the state labels, which is necessary due to the space limit. We have included a high-resolution version of the figure, and the reader can zoom in the figure to read the text. Different state annotations are also distinguishable by color, using the same coloring scheme as in Fig. 1.

Response to Review #3

Remarks to the Author:

The manuscript titled “diHMM: Multi-scale chromatin state annotation using a hierarchical hidden Markov model” by Dr Yuan and co-workers present an approach to infer chromatin states using histone marks. The manuscript is well written and the results suggest the method proposed may help in the determination of the higher-order features for chromatin state annotation.

Major concerns:

The main idea of diHMM is to capture chromatin states at two levels: 1. nucleosome-level information, which can be done with ChromHMM.

2. domain-level information, which can be experimentally demonstrated with Hi-C and computationally predicted by BLOCKS using BCP model. It would be great if the performance of diHMM is super on both levels, or at least on one.

Response: We do not claim that diHMM is superior than chromHMM at the nucleosome-level or than BLOCKS at the domain-level. These methods, among others, are useful for analyzing chromatin states, but they lose important information by focusing on a specific length scale. The main strength of diHMM is to provide a modeling framework to annotate multi-scale chromatin states simultaneously. Such multi-scale annotations are important for investigating the context-specific functions of enhancers and other elements.

We compared the performance of diHMM with chromHMM extensively in our original manuscript. In our view, the high degree of agreement between diHMM and chromHMM at the nucleosome level is actually a strength of our method, since it strongly suggests that coupling multiple length scales together does not introduce artifacts at the nucleosome level. This is important because the nucleosome-level annotations identified by chromHMM have been well-tested. The domain-level annotations identified by diHMM provide an additional layer of information that is important for biological functions of the chromatin.

The BCP model developed by Chen and colleagues is to identify local domains (called BLOCKS) with similar histone modification patterns. We compared the agreement between the segmentations obtained from these two methods. We found BLOCKS are typically two orders of magnitude broader than diHMM domains. Both BLOCKS and diHMM domain boundaries are significantly associated with the physical boundaries identified by Hi-C, but these two sets of boundaries are not significantly associated with each other. This was discussed in a new paragraph in the section entitled “Comparison of diHMM with existing methods”.

1. The author compared the overlaps between diHMM results and Hi-C interactions (Fig. S7). The enrichment of hubs for nucleosome-level enhancer states in diHMM in super-enhancer domains is in the range of 1.09 to 1.46. The numbers are quite low, and no statistical tests were provided.

Response: We thank the reviewer for his/her attention to details. We agree with the reviewer's assessment that the enrichment result is rather weak and noted this point in the manuscript. As suggested, we evaluated the statistical significance of these enrichments by using the Fisher-exact test, and found that all associations are statistically significant (p-values < 2.2e-16). This result is added in the revised manuscript.

2. diHMM predicts chromatin states at domain-level. The authors may want to make it clear whether diHMM can predict the interactions between domains and thus provides similar information as Hi-C for high-order interactions. If not, the authors may want to revise the line 45-47 in the introduction section.

Response: As discussed in the paper, there is significant association between diHMM domain-level states with various aspects of the physical interactions detected by Hi-C, but our method falls short of predicting genome-wide chromatin interactions. This has been clarified in the revised manuscript (in the section entitled “Context-dependent function of nucleosome-level states”).

3. Annotation of the chromatin states provided by diHMM relies on the prior knowledge about histone marks. The authors provided 30 nucleosome-level and 30 domain-level states. It would be very useful if authors can provide a clear guide to understand the biological functions of

these 30 states. i.e. at nucleosome-level, N1 and N2 were used to represent “Active promoter”. It seems that some active promoters lack of H3K27ac modifications.

Response: The annotation of chromatin states was manually assigned with the aim to maximize biological interpretability. To this end, we carefully examined three lines of evidence including: 1. the characteristic histone mark distribution of various functional elements; 2. the relative positions with respect to transcription start sites, exons, and transcription end sites; 3. the enrichment of known functional elements, such as CpG islands, exons, and repetitive elements. The results are summarized in Fig. 2 and S5. This procedure involves a lot of prior biological knowledge and is difficult to automate.

H3K27ac is well known to be associated with active enhancer (Creighton et al. PNAS, 2010; Rada-Iglesias et al. Nature, 2011), but it is not necessarily enriched in active promoters. In particular, it has been shown that H3K27me1 is often associated with active promoters (Ferrari et al. Mol Cell 2014). Since H3K27ac and H3K27me1 are mutually exclusive marks, H3K27ac is necessarily unenriched in these regions. In contrast, the most well characterized histone mark for active promoter is H3K4me3. Both N1 and N2 are enriched with H3K4me3 (Fig. 2), motivating our annotation. Furthermore, both N1 and N2 are strongly associated with TSS and CpG islands, further supporting the validity of our annotation.

4. What is the minimum data requirement on histone modifications for diHMM? The authors used 9 chromatin marks. If diHMM is very sensitive is to missing data, it will greatly limit the range of application.

Response: There is no minimum data requirement on histone modifications for diHMM. In fact, we can apply diHMM to analyze a single histone mark and obtain meaningful results. For example, we analyzed the H3K27me3 data in H1 by using two nucleosome-level states and three domain-level states. The domain-level states are particularly useful for identifying broad regions, which cannot be detected by traditional peak calling methods (Fig. S16). Missing data can be handled by using chromImpute (Ernst and Kellis, Nature Biotech, 2015), which has been successfully applied to impute epigenomic profiles in the Roadmap Epigenomics Project. The imputed data will be used as input for diHMM. In this revision we have added a few sentences in the Discussion section to address the missing data issue.

Minor issues:

1. A comparison between diHMM and BLOCKS would be desired.

Response: We have compared diHMM with BLOCKS as suggested by the reviewer. In particular, considering the efficiency, we trained a BCP model on 20kb resolution signal I on chromosome 17. The model identifies 25 BLOCKS, with average size of 3.2Mb. This is about two order of magnitude broader than an average diHMM domain, which is ~25kb. We then

tested whether the BLOCKS boundaries are associated with diHMM state transitions. Of 50 BLOCK boundaries, only 17 correspond to domain-level state transition. This overlap is not statistically significant (p -value = 0.06; Fisher's exact test). This is discussed in the section entitled "Comparison of diHMM with existing method").

2. To determine domain-level states, 20 consecutive 200 bp bins were used in the study. In Fig S8, the ranges of domain length vary significantly. A dynamic sliding window instead of 20 bins may provide a better algorithm.

Response: We thank the reviewer for the interesting suggestion. We assume the reviewer was referring to Fig. S2 instead of Fig. S8 (renamed as Fig. S9 in the revised manuscript) since the later was obtained by using chromHMM. In fact, we expect the length scale of different domains to vary, since they have different statistical properties and biological functions. For example, active promoters usually cover a few kilobases. In human, the length of a gene body varies from 1148 to 37.7 kb, whereas H3K27me3 associated repressive domains may span over 100kb (Ciabrell and Cavalli, Journal of Molecular Biology, 2015). Such biologically relevant variation would not be detected by using the sliding window approach suggested by the reviewer.

3. In Fig 3, the authors divided nucleosome-level enhancers into two categories: super-enhancer vs all others. It is not a surprise to see the striking differences between the two categories. It would be more reasonable to divide "strong enhancer" at nucleosome-level into two groups: super-enhancer vs not-super-enhancer for comparison.

Response: Reviewer 1 has a similar comment. Please refer to our response in the above.

Response to Review #4

Remarks to the Author:

Marco, Meuleman, Huang, Glass, Pinello, Wang, Kellis, and Yuan describe a new method for multiscale chromatin segmentation based on multivariate epigenomic profiling data, and the application of the method to analyze epigenomic data from three ENCODE human cell lines. They compared the results that they obtained from applying the new segmentation method with results that they obtained using a previously described single-scale chromatin segmentation method, ChromHMM. In the comparison, they used both gene expression data as well as predicted super-enhancer element locations (from an earlier study, referenced as [27] in this article, that was based on local enhancer density and H3K27ac ChIP-seq signal) for independent validation. The new chromatin segmentation method, diHMM, is based on a hierarchical hidden Markov model, with maximum-likelihood used for parameter estimation. The diHMM implementation used in this work incorporates both a nucleosome-level (200 bp bin size)

layer and a domain-level (4 kbp bin size) layer, though the general approach is not restricted to those specific bin sizes or layer types. The authors' key findings are that the use of diHMM (i) enables the detection of associations between specific nucleosome-level states and domain-level states, (ii) enables the discrimination, based on the local domain state (i.e., whether or not the local domain state is "super-enhancer"), of enhancer states that are more likely to be associated with higher levels of transcript abundance for the nearest gene; (iii) produces domain-level chromatin segmentations that are less fragmented and more biologically meaningful than those that are mapped using a single-scale HMM.

The work is highly original, timely, and would be of strong interest to a broad scientific readership in genomics, computational biology, and cell biology. The methodology and datasets used are, overall, appropriate to the problem. The overall statistical approach is judged to be sound, although two suggested statistical tests are included under 'minor issues/comments'. Overall, the conclusions of the article are well-supported by the described studies (only a few suggested additional pieces of information are listed under 'minor issues/comment'). The article appropriately cites previous work in the field. In terms of clarity, the article could use some minor improvements (specifically detailed under 'minor issues/comments') but there are no major issues with clarity, in this reviewer's opinion.

Response: We thank the reviewer for his/her positive evaluation of our work.

Minor issues/comments:

1. lines 55-56: "flexible modeling framework that systematically identifies combinatorial patterns at multiple length scales"-- it would be helpful to be a bit more precise here about the specific way in which diHMM improves over current popular approaches (vs. just enumerating positive attributes of diHMM). Are the authors claiming that diHMM is more flexible than previous approaches? More systematic? (evidence or an argument to substantiate that claim would be helpful in that case). It seems to this reviewer that the primary and most interesting way in which diHMM extends beyond previous approaches is that it gives a principled framework for detecting latent domain states and their associations with specific nucleosome-scale chromatin states.

Response: We thank the reviewer for the nice suggestion and modified the language accordingly.

2. line 77: Authors are commended for releasing the source code on GitHub. While this reviewer recognizes that research code is different from commercial software, readers may benefit from a bit of improvement to the code header comments and a bit of code clean-up (e.g., to remove "code introns" that have been commented out, or otherwise to explain their importance).

Response: We thank the reviewer for the suggestions and cleaned up the code on Github. .

3. line 82: Would be nice for authors to briefly share their thoughts/reasoning on the origin of the choice of "30 states" for nucleosomes and domains, especially for a non-expert reader.

Response: As discussed in our response to a similar comment by Reviewer 1, we used a pragmatic approach to select the number of chromatin states, aiming at using a relatively small number of states to capture the main variation of the signals. Similar approach has been widely used in previous studies (e.g. Hoffman et al, NAR 41, 827 (2013)). 30 states is somewhat arbitrary, but we found that the resulting states are biologically interpretable and that the results are quite producible by using other choices such as 20 and 25. Adding more states will substantially increase the numerical burden.

4. line 91, x-ref to Fig. 2a: are the data in Fig. 2a for all three cell lines, averaged? Clarification would be helpful, perhaps in the caption.

Response: This has been clarified as suggested. The emission probabilities shown in Fig. 2a are obtained after training the model on all three cell lines.

5. line 140, figure cross-reference "Fig. 1e": perhaps the authors meant "Fig. 2e" here?

Response: Yes, this is correct.

6. line 185, figure cross-reference "Fig. S8"; perhaps the authors meant "Fig. S2b and S8" here? Seems necessary to reference Fig. S2b as well as S8 in order to see the fragmentation.

Response: We have changed the text as suggested.

7. line 185 (Fig. S5 cross-ref): would be helpful to quantify the enrichment with some kind of overall test statistic.

Response: We have quantified the enrichment by using Fisher's exact test and found it to be statistical significant (p-value < 2.2e-16, Fisher exact test).

8. line 187 (Fig. S9 cross-ref): would be helpful to quantify the overall effect where it says "the trend was much weaker compared to diHMM", using some kind of test statistic

Response: We have not been able to find a suitable test statistic. To avoid confusion, we modified the sentence in the text to reflect that the average gene expression values are higher for diHMM than for ChromHMM.

9. line 233, "index i ": took this reviewer a bit of forward-reading to grasp that " i " refers to the 200 bp bin; would be good to clarify this.

Response: We have rewritten this part of the text for clarification.

10. line 243, " $e_k(b)$ ": maybe state that this is the emission probability? Then it is clear why the symbol " e " is being used. Also "depend only on nucleosome-level state" is a bit vague. I think what is meant is that the observations are independent of the domain state, conditioned on the nucleosome-level state.

Response: We have changed the text as suggested.

11. line 269: it would be helpful to introduce the scaling variable symbol s_i^n before it is used in the formula; the "meaning" of the symbol doesn't become apparent until line 272

Response: We have rearranged the text as suggested.

12. line 314: a bit of explanation as to why "chromosome 17" was selected would be helpful, as well as some comment on whether the maximum-likelihood model based on the chromosome 17 epigenomic data would be expected to approximately maximize the likelihood for the data on another chromosome. As a related question, how variable are the principal results if training data from a different chromosome were used?

Response: We initially tried to train our model genome-wide but was unsuccessful due to the numerical complexity. Therefore we arbitrarily selected a short chromosome (chr17) to train our model. We compared the results with those trained from chromosome 20, and found the results are highly reproducible (Fig. S2).

13. line 335: what if more than one gene is with +/- 2kb of the bin? Also, please clarify if log₂-transformed expression levels were converted to z-score (as would seem sensible), vs. non-log (i.e., "linear scale") expression levels.

Response: If more than one gene is within the +/- 2kb of the bin, then all of them will contribute to the average. The data we used were RMA normalized which involves log2 transformation.

14. lines 179-182: a bit more detailed explanation of precisely how (i) and (ii) differ, and what hypothesis is being tested by comparing them, would really help. So in (i), the k-center clustering is done on the ChIP-seq state information within a 4 kb block, and in (ii), the k-center clustering is done on the nucleosome states for the bins within the 4 kb block?

Response: We apologize for the confusion. In (i) we binarized the histone modification data at a 4kb resolution, and then applied chromHMM to the binarized data. This approach immediately results a 4kb resolution segmentation. In (ii), we used a two-step approach. First, we binarized the histone modification data at a 200bp resolution, as we did for diHMM, and applied chromHMM to obtain a 200bp resolution segmentation. Next, we combined the segmentation information within each stretch of 20 bins (so 4kb resolution) and searched for patterns by using k-center clustering. The second approach is conceptually more similar to our diHMM, but the two levels of chromatin states are not coupled.

15. Some overall quantification of how many 4 kb bins "switch color" between ChromHMM-4k and diHMM would be interesting to see; it is hard to guess from the heatmaps in Fig. S3c-d.

Response: Overall 63% of 4kb bins can be mapped to same color between ChromHMM-4k and diHMM. On the other hand, we have used a different method to evaluate agreement, which is quantified by a 'similarity score'. In the revised figure, the heatmaps represent correlation matrices. As such, the discrepancy does not correspond to the sum of off-diagonal entries.

REVIEWERS' COMMENTS:

Reviewer #2 (Remarks to the Author):

I thank the authors for thoroughly addressing the reviewer's comments. In my opinion, the paper has improved significantly in the review process.

Specific comments:

- Interestingly, in the comparison of the chr17 and chr 20 models revealed high consistency in the nucleosome-level states, but the models were less consistent in the domain-level states. I wonder whether the authors could comment on this.
- line 93: date -> data

Reviewer #3 (Remarks to the Author):

The authors have addressed all my questions.

Reviewer #4 (Remarks to the Author):

The revised manuscript all of the issues that I raised in the review. My previously stated positive assessment of the novelty, impact, and relevance of the work remains unchanged (see my comments to the original version of the manuscript).

My only suggestion would be: authors referenced Hoffman et al. in their response to my question about the rationale for 30 nucleosome-level and 30 domain-level states. It was a good response; perhaps the gist of that reasoning (with an included cross-reference to Hoffman et al.) could be included in the manuscript (line 88).

Response to Reviewer #2

Remarks to the Author:

I thank the authors for thoroughly addressing the reviewer's comments. In my opinion, the paper has improved significantly in the review process.

Specific comments:

- Interestingly, in the comparison of the chr17 and chr 20 models revealed high consistency in the nucleosome-level states, but the models were less consistent in the domain-level states. I wonder whether the authors could comment on this.

Response: We thank the reviewer for his/her appreciation of our effort and positive review. As suggested, we have added the following sentence commenting on our comparison results. 'Compared to the nucleosome-level states, the domain-level states are less robust, which likely reflects the smaller sample size in the training data.'

- line 93: date -> data

Response: We fixed this typo.

Response to Reviewer #4:

Remarks to the Author:

The revised manuscript addressed all of the issues that I raised in the review. My previously stated positive assessment of the novelty, impact, and relevance of the work remains unchanged (see my comments to the original version of the manuscript).

My only suggestion would be: authors referenced Hoffman et al. in their response to my question about the rationale for 30 nucleosome-level and 30 domain-level states. It was a good response; perhaps the gist of that reasoning (with an included cross-reference to Hoffman et al.) could be included in the manuscript (line 88).

Response: We thank the reviewer for his/her kind words and included the suggested citation in this revision.